# Legacy of Strength and Future Opportunities: A Qualitative Interpretive Inquiry Regarding Australian Men in Mental Health Nursing

**DOI:** 10.3390/nursrep15080287

**Published:** 2025-08-07

**Authors:** Natasha Reedy, Trish Luyke, Brendon Robinson, Rhonda Dawson, Daniel Terry

**Affiliations:** 1School of Nursing and Midwifery, University of Southern Queensland, Toowoomba, QLD 4350, Australia; natasha.reedy@unisq.edu.au (N.R.); brendon.robinson@health.qld.gov.au (B.R.); rhonda.dawson@health.qld.gov.au (R.D.); daniel.terry@unisq.edu.au (D.T.); 2Darling Downs Health, Toowoomba, QLD 4350, Australia; 3Centre for Health Research, University of Southern Queensland, Toowoomba, QLD 4350, Australia; 4Institute of Health and Wellbeing, Federation University Australia, Ballarat, VIC 3350, Australia

**Keywords:** mental health nursing, male nurse, workforce experience, professional fulfillment, camaraderie, teamwork

## Abstract

**Background/Objectives:** Men have historically contributed significantly to mental health nursing, particularly in inpatient settings, where their presence has supported patient recovery and safety. Despite this legacy, men remain under-represented in the nursing workforce, and addressing this imbalance is critical to workforce sustainability. This study offers a novel contribution by exploring the lived experiences, motivations, and professional identities of men in mental health nursing, an area that has received limited empirical attention. The aim of the study is to examine the characteristics, qualities, and attributes of mental health nurses who are male, which contributes to their attraction to and retention within the profession. **Methods:** A qualitative interpretive inquiry was conducted among nurses who were male and either currently or previously employed in mental health settings. Two focus groups were conducted using semi-structured questions to explore their career pathways, motivations, professional identities, and perceived contributions. Thematic analysis was used to identify key themes and patterns in their narratives. **Results:** Seven participants, with 10–30 years of experience, participated. They had entered the profession through diverse pathways, expressing strong alignment between personal values and professional roles. Five themes emerged and centred on mental health being the heart of health, personal and professional fulfillment, camaraderie and teamwork, a profound respect for individuals and compassion, and overcoming and enjoying the challenge. **Conclusions:** Mental health nurses who are male bring unique contributions to the profession, embodying compassion, resilience, and ethical advocacy. Their experiences challenge traditional gender norms and redefine masculinity in health care. Fostering inclusive environments, mentorship, and leadership opportunities is essential to support their growth. These insights inform strategies to strengthen recruitment, retention, and the future of mental health nursing.

## 1. Introduction

Mental health is increasingly recognised as a fundamental determinant of individual and collective wellbeing. It traverses physical health, social functioning, and economic productivity, representing a critical public health priority. The World Health Organization (WHO) [1] has identified mental health as a leading contributor to the global burden of disease, with significant implications for health care systems, workforce sustainability, and societal resilience. Despite this growing demand, the mental health sector continues to experience a chronic shortage of qualified professionals, particularly within nursing, a workforce central to the delivery of comprehensive, person-centred mental health care [2,3].

This dearth in the mental health workforce is further exacerbated by the persistent under-representation of men in nursing. In Australia, men constitute approximately 10% of the nursing workforce, a figure that has remained largely unchanged over the past two decades [4]. This trend mirrors global patterns, with WHO data indicating that men make up only a small proportion of the nursing workforce internationally, typically between 10 and 15% [1]. This gender disparity is particularly pronounced in mental health nursing, where the historical presence of men has diminished in the wake of broader sociocultural and educational shifts [5,6,7].

Although comprehensive longitudinal data is limited, historical accounts suggest a decline in male participation in mental health nursing, particularly following the deinstitutionalisation movement and the feminisation of nursing education and practice. While this under-representation reflects enduring gendered perceptions of nursing as a feminised profession, it also presents a strategic opportunity to address workforce shortages by enhancing the recruitment, retention, and professional development of men in mental health nursing roles [8]. Within this context it is also vital to acknowledge and challenge assumptions that may inadvertently reinforce traditional gender roles within mental health care [9,10,11].

While some narratives have historically associated men in mental health nursing with the management of patient aggression or physical interventions, such perspectives risks oversimplifying and potentially devaluing the diverse contributions that men provide to the profession, while also obscuring the multifaceted contributions of men in mental health nursing [9,10,11]. Contemporary mental health practice is focused on trauma-informed, relational, and recovery-oriented approaches that prioritise empathy, communication, and therapeutic relationships, skills, none of which are inherently gendered [12]. Nurses who are male should not be characterised by assumptions about physicality or confined to “gendered” roles that overlook their broader professional contributions but should be acknowledged for their capacity to foster trust, advocate for vulnerable populations, and lead with compassion and insight [9,10]. Their presence in mental health settings enriches the profession by challenging normative assumptions, diversifying perspectives, and modelling inclusive care [9,10,11]. Recognising and valuing these contributions is not merely a matter of equity; it is a strategic imperative for building a resilient, responsive, and future-ready mental health workforce.

For the purposes of this study, “male” refers to individuals who were assigned male at birth and who identify as men, recognising that gender is a complex and socially constructed experience. This terminology was chosen to reflect the gendered experiences relevant to our research focus while also acknowledging the complexity of gender as a socially constructed and individually experienced identity. We aim to use language that is both respectful and precise, avoiding terms that may reinforce stereotypes or suggest that men in nursing are inherently different from their peers.

## 2. Background

Historically, men played a foundational role in the evolution of nursing, particularly in psychiatric and institutional care [4]. Prior to the formalisation of nursing education, caregiving roles were often occupied by men in religious and military contexts [6,7,13]. Monastic orders such as the Alexian Brothers and the Knights Hospitaller provided care for those with mental illness as early as the 11th century, establishing a precedent for male involvement in mental health care [6,7]. By the 18th and 19th centuries, male attendants were a common feature of asylums across Europe and North America, often employed for their perceived physical strength and capacity to manage behavioural disturbances. These roles, while rudimentary and frequently lacking formal training, prepared the future professionalisation of mental health nursing in later years [5].

The institutionalisation of mental health care in the 19th century, particularly following legislative reforms such as the UK *Lunacy Act (1845)*, further entrenched male attendants and caregivers in psychiatric settings [5,14]. However, these roles were often stigmatised, poorly remunerated, and lacked professional recognition [15]. Not until the late 19th and early 20th centuries did formal training for mental health nurses begin to emerge, with varying degrees of accessibility among men. For example, in the US, male students were admitted to psychiatric nursing programs as early as the 1880s [16], while in Australia, formal training for mental health nurses who are male commenced in the early 20th century [17]. Despite these developments, the integration of mental health nursing into general nursing curricula did not occur until the latter half of the 20th century. Although a positive step forward, this contributed to the erosion of its distinct identity and the marginalisation of nurses who are male in this health care space [13,18].

The feminisation of nursing, both in practice and perception, has had profound implications for the recruitment and retention of men in the profession [13]. Cultural narratives that position nursing as inherently nurturing and, thus, aligned with traditional notions of femininity have contributed to the perception that men in nursing are anomalous or somehow deviant [19,20]. These gendered assumptions are particularly salient in mental health nursing, where emotional labour, therapeutic communication, and relational care are central to practice [13]. Consequently, men entering the field often encounter stigma, role ambiguity, and limited opportunities for advancement, all of which contribute to attrition and under-representation [4,8].

Nevertheless, empirical evidence suggests that men who choose to pursue careers in mental health nursing bring unique strengths and perspectives to the profession [20,21]. Studies have identified attributes such as emotional resilience, leadership, and a strong sense of justice as common among mental health nurses who are male [19,21]. Many men are drawn to the field through personal experiences of poor mental health, either directly or through family members, and are motivated by a desire to effect meaningful change in the lives of others [22]. Positive academic experiences, supportive clinical placements, and mentorship from experienced practitioners have also been identified as critical factors in shaping career trajectories [4].

Despite these motivators, mental health nurses who are male continue to encounter significant challenges. Workplace violence, gender-based role expectations, and a lack of institutional support are recurrent themes that are replete within the literature [4]. Men are often expected to assume physically demanding roles, particularly in crisis situations, without adequate training or recognition [23,24]. Moreover, the absence of targeted professional development pathways and the persistence of gendered stereotypes contribute to a sense of professional isolation and hinder career progression [4].

Addressing these challenges requires a multifaceted approach which includes educational reform, organisational change, and cultural transformation. Educational institutions must ensure that mental health nursing is presented as a viable and rewarding career path for all students, regardless of gender [4,23,25]. This includes integrating mental health content throughout nursing curricula, providing diverse clinical placements, and fostering inclusive learning environments. Health care organisations, in turn, must implement policies that promote equity, safety, and professional growth. Furthermore, it also includes mentorship programs, leadership development initiatives, and strategies to mitigate workplace violence and stigma [4].

Overall, due to a lack of targeted research regarding mental health nurses who are male [4], there is a pressing need for research that centres the experiences, not only to inform recruitment and retention strategies but also to enrich the theoretical and practical understanding of gender in nursing [8,26,27]. Despite some historical discussion of gendered dynamics in nursing, the specific experiences of men in mental health nursing remain underexplored. By showcasing the voices of men in mental health nursing, scholars and practitioners can challenge dominant narratives, disrupt gendered assumptions, and contribute to the development of a more inclusive and resilient workforce.

### 2.1. Problem Statement

Despite the historical prominence of men in mental health nursing, the current literature reveals a significant decline in their representation, particularly following the deinstitutionalisation of mental health care and the feminisation of nursing education [16]. While exact figures are limited, this trend has been documented recently [4]. Existing studies have examined general workforce challenges in mental health nursing, and few have focused specifically on the lived experiences, motivations, and professional identities of men who are nurses within this field. There is also a paucity of research that captures the self-perceived contributions among nurses who are men in mental health nursing, an area critical to understanding their professional value and role satisfaction. This gap is particularly concerning given the global shortage of mental health nurses and the potential to expand the workforce by engaging under-represented groups [4].

### 2.2. Aim

The aim of this study was to examine the characteristics, qualities, and attributes of mental health nurses who are male, which contributes to their attraction to and retention within the profession.

### 2.3. Objectives

In addition to the aim of the study, the objectives include the following:To identify the self-reported characteristics, qualities, and attributes that influence nurses who are male to pursue and remain in mental health nursing roles;To explore how mental health nurses who are male perceive their contributions to the recovery and wellbeing of individuals receiving mental health care.

### 2.4. Research Question

To address the aim and objectives the following research questions are sought:What are the self-identified characteristics, qualities, and attributes of mental health nurses who are male that influence their decision to enter and remain in the profession?How do they perceive their contributions to the recovery of individuals receiving mental health care?

## 3. Materials and Methods

To explore the characteristics, qualities, and attributes that draw men to and sustain their careers in mental health nursing, this study adopted a qualitative interpretive inquiry using focus groups [28]. This method was selected for its capacity to generate rich, in-depth insights through collective discussion and reflection [29]. By engaging participants in group dialogue, the study sought to uncover shared experiences, values, and motivations that influence men’s decisions to enter and remain in the mental health field [4]. The focus group approach also enabled the emergence of nuanced perspectives that may not be expressed within individual interviews, particularly in relation to gendered experiences within the profession [29].

### 3.1. Sample

Participants were purposively selected across Australia through known contacts and snowball sampling. Participants needed to be aged between 30 to 70 years of age and be within three distinct career stages. These career stages consisted of those who were retired mental health nurses, those currently employed in mental health nursing yet within seven years of retirement, and those who were in their early to mid-career as mental health nurses. Each focus group was planned to include between four and eight participants regardless of their career stages and sought to include up to twenty-four participants across the study. Recruitment was facilitated through professional networks, including those with long-standing ties, and through the distribution of promotional flyers via email, in person, and through various health services. Participants were required to identify as being male and have been assigned this gender at birth, have experience in mental health nursing, and be willing to participate in a focus group discussion.

### 3.2. Data Collection Tool

A semi-structured focus group guide was developed to elicit rich, narrative data from participants. The guide began with a broad, open-ended question: “Let’s talk about your experiences as a mental health nurse who is male by taking us back to when you first started in mental health nursing”. This question was designed to prompt reflection and storytelling. Additional prompts explored participants’ motivations for entering the profession, memorable experiences, challenges and rewards, influential role models, perceived contributions to the field, and reflections on the role of gender in mental health care (Appendix A). The guide was refined through internal review, along with input from mental health professionals, to ensure clarity and relevance. In addition to the focus group discussion, participants completed a short demographic questionnaire to provide background information on their age, years of experience, and employment history, which allowed greater time for discussion within the focus group.

### 3.3. Data Collection

Two focus groups were conducted among seven participants, where each participant attended only one of the two that were available. Sessions were held online via Microsoft Teams or a hybrid of face-to-face and online, which was dependant on participant availability, preference, and location. While Microsoft Teams allows participants to turn off their cameras, it was requested that cameras remain on throughout the session to maintain group interaction and dynamic. All participants agreed to this request and provided informed consent prior to the focus group. Each focus group lasted approximately 120 min. To ensure participant comfort and confidentiality, focus groups were facilitated by researchers who were not previously known to the participants. Co-investigators with potential prior professional relationships with the participants were excluded from the data collection to mitigate any risk of bias or perceived coercion. Three researchers (NR, TR, and DT) were present at each session to support facilitation and note taking. All sessions were audio and video recorded using Microsoft Teams, which complies with institutional data security protocols. Recordings were stored on encrypted, password-protected servers accessible only to the research team, ensuring data protection and confidentiality.

### 3.4. Data Analysis

Transcriptions were generated using Microsoft Teams’ automated transcription feature, reviewed, and amended collectively for accuracy by the researchers (NR, TR, and RD). Participants were invited to review their transcripts to confirm accuracy and were given the opportunity to amend, delete, or add to their responses. This process of member checking enhanced the credibility and trustworthiness of the data. Each participant’s data was de-identified and was assigned a code (e.g., P1, P2, P3, etc.) to ensure privacy and confidentiality was maintained.

Thematic analysis was employed to analyse the focus group data, following the six-phase framework outlined by Braun and Clarke [30], which includes the following:Familiarisation with the Data: Researchers began by immersing themselves in the data, reading and re-reading the interview transcripts to become deeply familiar with the content.Generating Initial Codes: Each data set was then assigned meanings, and significant quotes from the interviews were initially grouped by four researchers (NR, TL, BR, and RD). This involved systematically coding interesting features of the data across the entire data set and collating data relevant to each code. Rather than coding independently, the researchers engaged in a collaborative coding process. Through regular meetings, the team discussed and refined codes collectively, ensuring consistency and shared understanding. This consensus-based approach prioritised reflexivity and co-construction of meaning over formal interrater reliability metrics.Searching for Themes: The initial codes were subsequently categorised into potential themes independently by all researchers (NR, TL, BR, RD, and DT). This step involved sorting the different codes into themes and collating all the relevant coded data extracts within the identified themes.Reviewing Themes: The themes were further refined through collaborative discussions and consensus within the research team. This involved checking if the themes worked in relation to the coded extracts and the entire data set, generating a thematic map of the analysis.Defining and Naming Themes: Themes were named to capture their essence, using participant excerpts to enhance confirmability. This step involved ongoing analysis to refine the specifics of each theme and the overall story the analysis tells, generating clear definitions and names for each theme.Writing a report: The final step involved the selection of vivid, compelling extract examples, final analysis of selected extracts, relating the analysis back to the research questions and the literature, and producing a report of the analysis.

While inter-rater reliability was not calculated statistically, the collaborative nature of the analysis process ensured analytical rigour. Coding decisions were made through team discussions, supported by a shared codebook and detailed documentation of coding rationale, which enhanced the trustworthiness and transparency of the findings. Furthermore, triangulation of data sources was not employed due to the focused nature of this study; however, methodological rigor was supported through multiple validation strategies. Member checking was conducted to ensure the accuracy of participant transcripts, and collaborative coding among researchers enhanced analytical consistency.

To ensure reflexivity and minimise bias, co-investigators with potential prior relationships to the participants were not involved in data collection and throughout the data analysis process maintained reflexive journals or undertook reflexive discussions with the remaining researchers throughout the research process to monitor potential biases and strengthen the trustworthiness of the findings. These journals and discussions were used to document personal reflections and monitor potential influences on data interpretation. In addition, the research team also engaged in regular discussions to review emerging themes and ensure consistency and rigor in the analysis. While manual coding was used initially, Microsoft Excel software was employed to support data organisation and thematic development.

### 3.5. Ethical Considerations

Ethical approval for this study was granted by the University of Southern Queensland Human Research Ethics Committee (#ETH2023-0593). Informed consent was obtained from all participants prior to data collection. Participants received a detailed information sheet outlining the purpose of the study, the voluntary nature of participation, and their right to withdraw at any time without consequence. Confidentiality and anonymity were assured through the de-identification of all data. All audio and video recordings and transcripts were stored securely in password-protected cloud storage in line with ethical approval.

## 4. Results

A total of seven mental health nurses who are male participated in the study, and they included a diverse cross-section of mental health nurses who are male who ranged in age from 33 to 67 years. Their backgrounds reflected a wide variety of previous occupations, from station hand, farmer, boiler maker, and bank teller to various positions in health care. Most men began their careers as registered nurses before transitioning into mental health nursing. The age at which they commenced mental health training ranged from 21 to 30, indicating both early and late career transitions. Training pathways were also varied, with the majority completing hospital-based mental health nursing certificates due to training available at the time, while one participant pursued university-level education (Table 1).

The interviews revealed a rich tapestry of themes that reflect the lived experiences, values, and challenges of mental health nurses. These themes encompassed mental health being at the heart of health, being shaped by the influence of experiences, along with personal and professional fulfillment. This is followed by camaraderie and teamwork, which were highlighted as essential to sustaining the emotional demands of the profession. The theme of profound respect for individuals and compassion encompassed core strengths such as empathy, emotional intelligence, person-centred care, and advocacy, extending also to interdisciplinary respect. Participants also identified significant challenges, including workplace violence, stigma, and emotional burden, while also overcoming these. Each of these themes offers insight into the complex, relational, and evolving nature that men were experiencing in mental health nursing and are discussed in detail.

### 4.1. The Heart of Health

The first theme, or most prominent theme, that emerged was that being a mental health nurse was profoundly a human and relational profession, and this was indicated among all participants, and it was suggested to often remain misunderstood both within and outside the broader health care ecosystem. For nurses who are male, mental health nursing was said to offer not only a career but a space to forge a meaningful identity, one that is grounded in emotional intelligence, personal growth, and a deep commitment to the wellbeing of others.

Participants revealed they arrived in the profession through diverse and often unconventional pathways. Some are drawn by personal or familial connections to poor mental health, others by chance encounters, or some by the search for more meaningful work after careers in unrelated fields. Many wanted to be part of the solution of the negative way patients with poor mental health were treated by health care workers observed in general nursing. Regardless of how they entered, there was a shared sense of purpose that emerged through their experiences. It was in these moments that they discovered a sense of alignment between their personal values and the professional role. Mental health nursing “allows us to look holistically at the person” (P3), one nurse explained, highlighting the depth of care involved.

Overall, what emerged among participants was a quiet but powerful assertion of identity. Mental health nurses who are male highlighted they were not defined by the stereotypes imposed upon them. Instead, they defined themselves through their actions, their integrity, and their commitment to making a difference. “I still love mental health nursing… it’s afforded me lots of opportunities,” P4 shared, reflecting the enduring value of the work. In this way, mental health nursing was indicated to be more than a job, a space where men can embody a different kind of strength, one that is thoughtful, grounded, and deeply human.

### 4.2. Personal and Professional Fulfillment

All participants expressed a strong sense of satisfaction with their career journeys, highlighting that mental health nursing offered unique opportunities not typically available in general nursing. One participant reflected:


*Psychiatric nursing has given me that opportunity to do different roles and do different things…I doubt if I would have had the opportunities that I’ve had if I had of stayed in general nursing. (P3)*


Another participant emphasised the value of career progression and financial recognition when stating that there were


*Wonderful opportunities and career and progression that I’ve been offered, and the decent pay to get there, which is nice. (P2)*


Several participants spoke about their involvement in policy and cultural change within their organisations. One described a career highlight involving the development of a “rural regional health service” that integrated Aboriginal mental health workers, advocating for their recognition and equal employment status. Another participant, who became a Nursing Unit Manager, shared how they led a cultural shift focused on discharging patients from a secure unit:


*Whilst it was a real highlight for my career, it was certainly done with an amazing team of people and… in a three-year period, we discharged 40 people… In a secure unit that was just like, wow [unheard of], you know… there was a lot of job satisfaction (P7).*


This participant also noted that management was a particularly fulfilling phase of their career, as it allowed them to implement change and introduce innovative ideas.

Two participants currently work as nurse educators and described this role as a significant and rewarding career milestone. One called it the most fulfilling work they had done, while the other reflected on the dual nature of the role:


*Working with… student nurses… feels like your experience is going towards a future legacy of nurses…*


later adding that it was also concerning, as they once saw


*…300 people coming into the [mental health] grad program in Queensland [but] is now about 120 people. (P7)*


The majority of participants had long-standing careers in mental health nursing and had witnessed significant systemic changes, particularly the shift from the institutionalised medical-model of care to a more community-based, holistic approach. One participant described the satisfaction of helping establish a “Living Skills Centre” with a rehabilitation focus and also starting a “Community Care Unit”, which enables the support of patients’ in their own environments.

Another participant highlighted the learning and fulfillment that came with community-based care:


*…getting out of the more acute environments and getting into community. I got into a role where I could start learning more psychotherapy skills, which was amazing to finally get to do what I felt as to be actually helping people. (P2)*



*These diverse personal and professional journeys highlight the rich, rewarding nature of mental health nursing. Whether motivated by personal values, role models, or a desire to drive meaningful change, participants found opportunities for growth, leadership, and lasting impact.*


### 4.3. Camaraderie and Teamwork

In addition to the personal and professional fulfillment found in career progression, participants also emphasised the importance of camaraderie and teamwork. This culture was not incidental but a foundational element of professional identity and practice. Participants illustrated how collaborative dynamics, mutual respect, and shared emotional labour contributed to both individual wellbeing and collective efficacy within the various health care settings where they were or had been working. In addition, participants described a professional environment where status distinctions were minimised, fostering a sense of equality and shared purpose. One participant reflected on their early experiences in mental health nursing, stating,


*There was no delineation between everyone… it didn’t matter if you were a medical superintendent or the groundsman on the mower, everyone was on the same level [where each person was treated with the same respect and value regardless of their role or status]. (P1)*


This egalitarian ethos was described in contrast to general nursing, with participants reporting that this structure within mental health was associated with open communication and a sense of mutual support among team members.

Participants also highlighted the importance of informal social interactions in building and sustaining camaraderie. Shared interests and humour were frequently cited as mechanisms for establishing rapport and reinforcing team bonds. Participant, P1, noted the value of “banter” in building relationships with clients and colleagues, describing it as “a handy tool” that contributes to rapport and mutual understanding. Similarly, P4 recounted the formation of a staff soccer team, which served both as a morale booster and a community-building initiative: “residents in the hospital… come out and watched the game” (P4). These examples highlighted that informal, non-clinical interactions were associated with the development of professional relationships and a positive team dynamic.

Another significant element of camaraderie and teamwork was the role of mentorship and collective learning in fostering a supportive team culture. Participants describe environments where knowledge was shared freely, and professional development was encouraged through collaborative practice. One participant spoke about the satisfaction derived from mentoring new nurses, stating, “some of the highlights would be working with students, precepting them… investing in the new psychiatric nurses” (P4). This investment in others was then mirrored by a reciprocal learning process, where insights were gained from colleagues and clients. This participant added, “I’ve learned so much from clients. It’s unbelievable” (P4), highlighting the dynamic and dialogical nature of mental health care.

Emotional resilience, supported by strong team dynamics, was another critical aspect of participant narratives. The work they do was frequently described as emotionally demanding, while acknowledging the psychological toll of patient care, particularly in the context of loss and trauma. One participant reflected on the emotional volatility of the profession, noting, “it’s a job that can be full of stress… if we talk about loss of clients through suicide” (P4). However, the presence of a “good, supportive team” (P4) was consistently cited as a mitigating factor and providing emotional support during all challenging times. Another participant similarly described the workplace as one characterised by “a lot of support and nurturing… just being part of the team” (P6), reinforcing that emotional wellbeing remained closely tied to the quality of interpersonal relationships within the team.

The significance of teamwork was perhaps most vividly illustrated in moments of crisis. For example, a participant recounted a violent incident involving a patient, during which colleagues responded swiftly and effectively. He stated it was the “good staff, there in the nurses’ station… [that] came to my assistance” (P5). This rapid, coordinated response not only ensured physical safety but also reinforced a sense of trust and reliability among team members. Such experiences highlighted the critical role of teamwork in maintaining both clinical safety and psychological security.

### 4.4. Profound Respect for Individuals and Compassion

Beyond the strength of teamwork and shared responsibility, participants also spoke of a deeper, more personal dimension of their practice, one rooted in profound respect for individuals and compassion. This theme was deeply embedded in the narratives among participants, encompassing empathy, emotional intelligence, non-judgmental attitudes, person-centred care, and forming meaningful therapeutic relationships. These qualities were not only foundational to impact mental health nursing but also reflect a deep-seated respect for the personhood of everyone, including colleagues across disciplines. One of the most vital aspects was the emphasis on building and maintaining therapeutic relationships. One participant shared a powerful story of working with a client over three years, highlighting the transformative power of simply being present and understanding the client’s worldview:


*She didn’t attempt suicide again, ever, and her substance use declined substantially… I attribute that to nothing else than developing a relationship, just spending time with her, wanting to see the world through her eyes. (P2)*


This statement exemplifies the profound respect for the individual’s lived experience and the compassionate commitment to long-term, person-centred care. In addition, another participant reflected on the emotional intelligence required in mental health nursing, stating:


*If somebody asked me point blank what’s the first criteria I look for in my staff, I look for emotional intelligence. (P3)*


This highlights the recognition that technical skills alone are insufficient, but the ability to empathise, listen, and respond with sensitivity is paramount. Emotional intelligence enables nurses to navigate complex emotional landscapes, advocate effectively, and build trust with clients and colleagues alike.

It was also revealed that a strong non-judgmental and holistic approach to care was paramount. One nurse, P4, described the importance of being “genuine”, “transparent”, and “valuing their opinion,” which are essential to creating a calm and dependable environment. This approach fosters trust and safety, allowing clients to open and engage in their care. P4 also emphasised the importance of “being patient, not overbearing, being person-centred,” which reflects a deep respect for the autonomy and dignity of everyone. One participant described the simple yet profound act of taking someone out for a smoke as a way of listening and creating a low-stimulus environment. This act, though small, reflects a deep understanding of individual needs and a commitment to meeting them with empathy and presence.

The findings also reflected a profound respect for interdisciplinary collaboration. Nurses often act as intermediaries between the medical model and the client, translating complex information and ensuring that care remains person centred. One participant described this role as navigating


*between that model [medical model] and directly with the client and their family… the ability to feel trusted in our relationship, that a client would trust you with the information they give you. (P4)*


This trust demonstrates the respect they show for both the client and their professional colleagues. Several participants shared stories of role models who embodied compassion and creativity in their practice. These individuals were described as “highly intelligent”, “creative,” and “*real time pacifists would exhaust all efforts to try to avoid using things like physical[ly] restraining somebody… sometimes to his own detriment*” (P5). These findings illustrate the depth of commitment to non-coercive, respectful care, even in challenging circumstances.

### 4.5. Overcoming and Enjoying the Challenge

The final theme explores the challenges being experienced and the tenacity of mental health nurses who are male to rise to the challenges and overcome them. Overcoming the challenges they faced brought a sense of satisfaction and accomplishment that permitted their clinical practice to align more closely with their value system of raising dignity and respect for consumers and their colleagues.

“There’s a stigma attached with mental health (consumers) and the staff that work with them.” (P3) This statement, among many others in a similar vein, indicated the participants held a sense of natural justice spurred from an empathic response to people who they considered were treated or appraised by others as less dignified. Examples of their recollections of stigmatising comments and actions raised by colleagues and community members showed evidence of an empathic response prompting advocacy for the aggrieved, sometimes steering their career pathway towards mental health nursing to become a part of the solution. P1 lamented his colleagues:


*ignorance and disregard for mental health clients…. [and] relegating (a mental health consumer) to a room at the end of the ward—where basically they were left alone, staff did not want to spend time with them. (P1)*


P3 recalled an incident witnessed of a mental health consumer experiencing stigma directed at them by a colleague when they stated,


*It’s almost depressing at times, but you get a lot of… people in your group who’ll say, well, you know, they’re [patient] a waste of time, aren’t they?… and I think that while it’s improved, we’ve got a long way to go and, look, hopefully we’ll get there in the future. (P3)*


Stigma towards mental health nursing as a career option also challenged the participants. They indicated that mental health nursing was perceived by colleagues as being “cushie” (P5), “dumbed down” (P3), “shunned” (P3), “not well recognised or well regarded” (P1), and “lazy, incompetent” (P3), with the specialty in decline (P4).

A third stigma experienced by the participants was “gender discrimination in terms of expectations that nurses are predominantly female.” (P2) This stigma was expressed by patients who had told P1 “it was wrong males are nurses” and community members who assumed that “‘Oh, you’re a male nurse? You must be gay?’” (P1). Notably, the stigma experienced by the male mental health nurse participants was often internalised as a motivator for them to be part of a system that could change the stereotype. The challenges experienced by stigma prompted the participants to rise to the challenge. Regarding the stigma witnessed from a colleague to a consumer when one participant stated:


*I thought we’ve got to be able to take better care of people than that, and there must have been a reason why this lady had done, you know (self-injurious behaviour). Those kinds of incidents spurred me on as well. (P3)*


Also, in response to being targeted with stigma for being a nurse who was male, “I just leant into it, because it didn’t bother me at all” (P2) and “so it’s funny, it’s always been proposed to me like it’s a disadvantage to be a male in this situation, but I don’t feel that at all.” (P2).

The emotional burden of mental health nursing was reported as a challenge, particularly when after investing their time and energy into a consumer, exhausting their efforts and resources, a negatively experienced result still eventuated. Suicide, restrictive practices, disclosures of trauma from consumers, and physical and psychological injuries to colleagues were perceived as traumatic. Participants reported the emotional burden as taxing, with some colleagues leaving employment and their profession due to the toll taken by various incidents. Aggression, violence, and complex presentations in mental health nursing were listed as further challenges, which sometimes became overwhelming, with a participant stating, “you get sick of the violence” (P1).

However, resilience among participants through intense incidents was evident when one participant stated,


*I’ve certainly been involved in hundreds of fairly significant events, but umm, there was, I guess a great feeling of camaraderie and debriefing that, none of those incidents really come to the forefront for me. (P7)*


Stories expressed by the participants indicated their ability to maintain a therapeutic alliance during aggressive incidents such that “when his mood came down, and stabilized, he went around and just thanked everybody” (P4). In addition, another shared a similar experience when stating,


*What I do remember is when he got better, which he did, and he was about to leave the unit, he came up and he personally thanked me. He said. Thank you very much for not hurting me and I said, look, we’re not here to hurt anybody. (P3)*


P6 identified that working with people “on their worst day” can be rewarding “if you listen or, you know, some sort of intervention” that leads to a positive outcome.

Where participants encountered challenges, their ability to respond with resilience and purpose emerged as a unifying theme. When current practices contributed to the stigma experienced by consumers, participants responded with advocacy, compassion, and non-judgmental care to uphold dignity and respect. This commitment often translated into efforts to challenge and improve systemic practices, aligning their work with deeply held values. As one participant reflected, “that’s what sort of makes you think that your work is worthwhile… at least people would matter” (P4). Others acknowledged the emotional complexity of the role: “It’s a job that can be full of highlights. It’s a job that can be full of stress” (P3); “It can test your own innate capacities to relate to somebody and interpret what’s going on for them and try to be a bit of an assistance” (P4). One participant powerfully summarised the emotional depth of the role: “I’ve walked alongside on their terrifying, scary, horrific, but also inspiring stories and lives” (P2). These reflections illustrate how mental health nurses who are male transform adversity into meaningful practice, reinforcing their identity as compassionate professionals committed to ethical care and systemic change.

## 5. Discussion

Overall, the study reinforces that mental health nursing is a uniquely relational and emotionally demanding profession, one that is often misunderstood within the broader health care system [31,32]. Mental health nursing offers nurses who are male an opportunity to construct a meaningful professional identity. This career pathway is grounded in emotional intelligence, personal growth, and a commitment to others’ wellbeing, aligning with previous research that highlights the therapeutic use of self and the centrality of interpersonal connection in mental health care [4]. The diversity of entry pathways, ranging from personal experiences with mental illness to career changes, reflects the profession’s capacity to attract individuals seeking purpose-driven work [21,33,34].

A defining feature of the role, as described by participants, was advocacy, not only for clients but also for the profession itself, echoing the literature that identifies mental health nurses as key agents in challenging stigma and promoting patient rights [35,36]. However, participants also reported facing persistent stigma and gendered assumptions, such as being perceived as less competent or stereotyped based on their gender, which mirrors the findings from studies on male nurses’ experiences of marginalisation in feminised professions [4,37]. Despite these challenges, the narratives revealed a strong sense of professional pride and identity, with participants describing their work as deeply human and transformative that challenges traditional notions of masculinity and redefines what it means to care [4,37].

The findings also revealed mental health nursing offers a wide range of career pathways that contribute to long-term retention. Participants described entering the profession through diverse routes, some from general nursing, others from unrelated fields, yet all found alignment between their personal values and professional roles. Many spoke of career satisfaction, leadership opportunities, and the ability to influence policy and practice. These insights align with the recent literature that highlights the importance of meaningful work, role diversity, and visible role models in supporting recruitment and retention in mental health nursing [33,38,39]. Importantly, participants remained in the profession despite its challenges and due to the meaning they gained from overcoming those challenges. Their stories reflect a deep commitment to advocacy, ethical care, and systemic change, factors that have been shown to enhance resilience and professional longevity within nursing and other care industries [40,41,42].

Camaraderie and teamwork provide a crucial backdrop for understanding how nurses who are male navigate and reconstruct masculine identity in their professional roles. The literature has increasingly recognised that collaborative, egalitarian environments, where emotional labour is shared and hierarchies are minimised, can foster psychological safety and mutual respect [33,34,43]. In such settings, informal interactions, mentorship, and collective resilience are not only protective against burnout but also create space for alternative masculinities to emerge [8,44]. Rather than reinforcing traditional ideals of stoicism and control, these team dynamics support expressions of empathy, vulnerability, and relational strength [8,44]. This context enables nurses who are male to embody a form of masculinity that aligns with the core values of mental health care, compassion, presence, and ethical engagement, setting the stage for a broader redefinition of what it means to be a man in nursing [44].

Within this collaborative and emotionally supportive environment, participants’ experiences further illuminated how mental health nurses who are male negotiate and redefine masculinity in ways that align with the relational and ethical values of the profession [4,8]. Building on this foundation, the study illustrates how mental health nurses who are male actively negotiate and redefine masculinity through values such as compassion, emotional intelligence, and therapeutic presence [27]. While the participants did not always explicitly discuss gender, their reflections offer a compelling insight into how being a man in mental health nursing intersects with the core values of the profession. Traditionally, masculinity has been associated with traits such as emotional restraint, authority, and control [37]. However, the participants challenged these norms by embracing empathy, patience, and vulnerability, which are qualities essential to building meaningful therapeutic relationships, reflecting a compassionate masculinity that values connection over control [4,45].

Furthermore, there was a prioritisation of emotional attunement over technical dominance that was demonstrated, which reflects a shift in how nurses who are male define professional competence and personal strength [4,45]. In addition, the findings also highlighted how nurses who were male acted as advocates and intermediaries, navigating institutional power structures to support clients [46]. This role requires assertiveness and courage, which are traits often linked to masculinity [44], but among these, men in the care industry were able to use such approaches in service of justice, respect, person-centred care, and upholding client dignity [4,8].

Participants also described how stigma, whether directed at clients, the profession, or themselves, was not only a challenge but a motivator. Rather than deterring them, these experiences deepened their commitment to advocacy and ethical care. This aligns with recent research showing that men in caring professions often reframe stigma as a call to action, using it to construct a compassionate performative masculinity [8,44,47]. Also called ethical masculinity, this is a version of masculinity that is centred on empathy, relational care, and moral responsibility [8,44,47]. Rather than distancing themselves from the emotional labour traditionally associated with nursing, these men actively embrace and perform compassion as a strength, aligning their professional identity with values of integrity, service, and social contribution.

Moreover, several participants described their roles as change agents, leading cultural shifts, advocating for marginalised groups, and influencing policy. These actions reflect a broader conceptualisation of nurses as leaders in systemic transformation [48]. Such leadership is not rooted in hierarchy but in relational influence and ethical engagement, echoing contemporary models of participatory leadership in health care [49].

Furthermore, several participants referenced other male role models who were nurses before them, who were pacifists, creative, and deeply committed to non-coercive care. These figures model a form of masculinity that remains ethical, emotionally present, and intellectually engaged, which was from the rigid, hierarchical models often associated with traditional male roles in health care [4,33]. Overall, being a man in mental health nursing involves redefining masculinity through the values of compassion, respect, and relational depth. These men are not only caregivers but also cultural change agents, demonstrating that strength lies in empathy and leadership in listening [4,8,50].

### 5.1. Limitations

This study has several limitations that should be acknowledged. First, the small and homogenous sample, comprising only seven participants, limits the generalisability of the findings to broader populations and different cultural and health care contexts. The use of focus groups, while valuable for generating rich, collective insights, may have introduced social desirability bias, where participants conformed to dominant views or withheld more vulnerable perspectives. Additionally, the reliance on retrospective self-reporting may have affected the accuracy of participants’ recollections. Although reflexive journals and member checking were employed to enhance credibility, manual coding and interpretation by researchers with professional backgrounds in mental health may have introduced interpretive bias. The online and hybrid format of the study, while necessary for accessibility, may have constrained natural interaction and depth of discussion. Furthermore, recruitment through professional networks may have resulted in a sample biased toward individuals with more positive or engaged perspectives, potentially excluding those who are disillusioned or have left the profession. Finally, with only two focus groups conducted, there is a possibility that thematic saturation was not fully achieved.

### 5.2. Implications for Practice

The findings of this study have several important implications for both mental health nursing practice and higher education. First, the diversity of career pathways and the strong sense of purpose expressed by participants highlight the need for recruitment strategies that emphasise the relational, ethical, and transformative nature of mental health nursing. Campaigns and outreach efforts should showcase male role models and the profession’s alignment with values such as advocacy, emotional intelligence, and systemic change.

In clinical practice, the emphasis on camaraderie, emotional resilience, and egalitarian team dynamics suggests that workplaces should foster psychologically safe environments where emotional labour is shared and alternative masculinities can thrive. Structured mentorship programs, peer support initiatives, and reflective practice groups may enhance retention and wellbeing among male nurses.

For higher education institutions, the study highlights the importance of embedding gender-sensitive and emotionally intelligent pedagogy into nursing curricula. Educators may create space for students to explore identity, challenge stigma, and develop therapeutic presence. Given the appreciation for career progression and leadership opportunities among participants, academic programs should also highlight diverse career trajectories within mental health nursing, including roles in education, policy, and community-based care. Finally, the findings call for a redefinition of professional competence, one that values compassion, presence, and advocacy as much as technical skill. This shift has the potential to reshape how nursing is taught, practiced, and perceived, particularly for men entering a traditionally feminised profession.

## 6. Conclusions

This study offers valuable insights into the experiences of male mental health nurses, revealing a profession deeply rooted in relational care, emotional intelligence, and ethical advocacy. Mental health nursing is not only as a career but as a space for personal growth, identity formation, and meaningful contribution, along with resilience, compassion, and a commitment to challenging systemic inequities. It remains important for supportive team environments, diverse career pathways, and opportunities for leadership and systemic change. These elements not only sustain long-term engagement but also redefine traditional notions of masculinity within health care. The study highlights the need for practice environments and educational programs that foster emotional attunement, mentorship, and reflective practice. Ultimately, mental health nurses who are male emerge as cultural change agents, professionals who embody a compassionate masculinity that values connection over control and advocacy over authority. Their stories challenge outdated stereotypes and offer a compelling vision for the future of mental health nursing, one that is inclusive, value driven, and transformative. These insights have important implications for recruitment, retention, and the ongoing evolution of nursing education and practice.

## Figures and Tables

**Table 1 nursrep-15-00287-t001:** Demographics of participants.

Participant Number	Current Age	Previous Work History	Was Registered Nurse First	Mental Health Nurse Career Start	Age at Training	Type of Training	Workplace Setting
P1	56	Station Hand	Yes	1998	30	Hospital-Based Mental Health Nursing Certificate	Regional hospital setting with inpatients
P2	33	Retail, Art Therapist, and Psychotherapist	Yes	2019	28	University Masters of Mental Health	Metropolitan and regional community setting with outpatients
P3	52	Farmer	Yes	1997	25	Hospital-Based Mental Health Nursing Certificate	Regional hospital setting with inpatients
P4	65	Registered “Mental Handicap” Nurse and Remote Area Nurse	Yes	1982	21	Hospital-Based Mental Health Nursing Certificate	Rural hospital setting with inpatients and outpatients (retired)
P5	67	Nil Recorded	No	1981	24	Hospital-Based Psychiatric Mental Health Nursing Course	Regional hospital setting with inpatients (retired)
P6	53	Bank Teller and Enrolled Nurse	Yes	1999	28	Hospital-Based Mental Health Nursing Course	Metropolitan hospital and community setting with inpatients and outpatients
P7	60	Boiler Maker	No	1993	26	Hospital-Based Psychiatric Mental Health Nursing Course	Regional hospital setting with inpatients

## Data Availability

The data presented in this study is available on request from the corresponding author due to requiring ethical clearance to share the data.

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
