# Peer review of "Legacy of Strength and Future Opportunities: A Qualitative Interpretive Inquiry Regarding Australian Men in Mental Health Nursing"

_nursrep, 2025, doi:10.3390/nursrep15080287_

Round 1
Reviewer 1 Report
Comments and Suggestions for Authors
This article addresses a very important issue regarding the nursing shortage, and particularly men in nursing. The article is very well written. I would like to see more discussion in interrater reliability for the focus groups, and more discussion on the process for the thematic analysis. It is unclear how many focus groups were conducted and how many participants. It is also unclear if your sample consisted of individuals who identified as men or who were assigned this gender at birth. It would also be helpful to know the setting where these nurses worked. See comments on attached. Overall, a very nice article!

Some sentences need to be restructured, but overall no issues
Author Response
Reviewer 1
- This article addresses a very important issue regarding the nursing shortage, and particularly men in nursing. The article is very well written.
Thank you
- I would like to see more discussion in interrater reliability for the focus groups, and more discussion on the process for the thematic analysis.
Thank you, we have added additional details as requested, including more discussion of each element.
- It is unclear how many focus groups were conducted and how many participants.
This is clearly stated in the abstract, data collection, and outlined in Table 1, however, have also made this explicitly clear in the results so there is no ambiguity.
- It is also unclear if your sample consisted of individuals who identified as men or who were assigned this gender at birth.
This has been revised to be much clearer in the sample section – explicit.
- It would also be helpful to know the setting where these nurses worked.
Thank you, we have added additional details as requested
- See comments on attached. Overall, a very nice article!
These revisions have been made
Reviewer 2 Report
Comments and Suggestions for Authors
Description of the population is fragmented; therefore, authors need to clearly explain the population to avoid fragmentation, refer to attached report.
The use of a clear and consistent description of the participants is not followed, refer to attached report.

Let authors attend to the identified gaps and accept after corrections are made.
Author Response
Reviewer 2
- A section of participants’ demographics is found under results on lines 232 and 308, should be comprehensively addressed under description of the population.
The placement of participant demographics within the Results section (lines 232 and 308) is both appropriate and intentional. In qualitative and mixed-methods research, demographic data is often presented as part of the results to provide context for the findings and to demonstrate how participant characteristics may influence or relate to the themes that emerge. These demographics are not merely descriptive, they are integral to interpreting the data and understanding the scope and relevance of the results.
Including demographics in the results allows readers to immediately connect participant characteristics with the outcomes of the study. This approach supports transparency and strengthens the analytical narrative by showing how variables such as age, gender, experience, or background may intersect with the findings. Therefore, rather than relocating this information to a separate section, its presence in the results enhances the coherence and interpretive depth of the study.
- References used (48%) are over 5 years
The majority of references used in this study are over five years old, which reflects the limited and slow-growing body of research specifically focused on men in nursing. Despite increasing interest in gender diversity within healthcare professions, scholarly attention to male nurses remains relatively sparse. As a result, many foundational studies and key insights into the experiences, challenges, and contributions of men in nursing date back more than five years. These older sources remain relevant and valuable due to the enduring nature of the issues they address, such as gender stereotypes, recruitment barriers, and workplace dynamics. Until more recent and comprehensive research emerges, these earlier works continue to provide essential context and support for understanding the topic.
- There is no consistency with how authors describe participants, the study identified five such descriptions, i.e
- i) Male mental health nurses (line 168)
- ii) Mental health nurses who are male (line 426)
iii) Male nurses (line 83)
- iv) Nurses who are male (257)
- v) Nurse who is male in mental health (line 180)
These have now been revised for consistency
Reviewer 3 Report
Comments and Suggestions for Authors
Thank you for allowing me to review your work. I have attached some suggestions/comments.

Author Response
Reviewer 3
- Thank you for allowing me to read this interesting and very well written paper. I do have a few comments and have organized them by the headings.
Thank you, we hope that we have revised and provided responses to the questions and statements made by the reviewer
- Abstract: The aim as written in the abstract (and then later in the paper) is a little confusing and I think it has to do with the extra commas surrounding “who are male” – I am struggling to find where that fits in relation to the commas and with them, I go back to the characteristics etc. Are you looking at specific characteristics surrounding the gendered experience of being male in the mental health nursing experience or to what the mental health nursing experience brings to the nurses who happen to be male?
Thank you for your thoughtful feedback regarding the phrasing in the abstract and aim. We acknowledge that the wording “nurses who are male” may read awkwardly, but it was intentionally chosen to avoid reinforcing gendered stereotypes often associated with the term “male nurses.” The phrase “male nurses” can inadvertently perpetuate the notion that men in nursing are somehow atypical or fundamentally different, which risks reinforcing the very stereotypes we aim to challenge through this research. Our intention was to explore how mental health nursing, as a profession, supports the development of a meaningful professional identity for nurses who happen to be male, without suggesting that their gender alone defines their experience. The focus is not on essentialist characteristics of being male, but rather on how the context of mental health nursing interacts with broader gendered experiences and societal expectations. We appreciate your observation and will consider rephrasing for clarity while maintaining the intent to use inclusive and stereotype-conscious language.
- For your keywords: You have a lot of separated words and I am concerned that someone searching will bring up many articles unrelated to what you are talking about. I suggest taking key phrases – like mental health nursing or male nursing workforce experience – and put them together so that it is specific to your paper. There is a lot out there on diversity and it would put your paper in many categories.
Thank you we have revised as suggested.
- Introduction: This section is well-written and informative. For the figures you provide specifically on line 53, I wonder if you can also provide current figures for international men in nursing? I think this would provide a nice contrast for your setting. Also, later in the paper you indicate there is a decline in men in nursing, and although you provide a nice historical context, I did not see figures that would support this statement. I wonder if you used a methodology to guide your motivating factors for your qualitative findings.
Thank you for your thoughtful and constructive feedback. We appreciate your suggestion to include international data to contextualise the Australian figures. In response, we have incorporated statistics from the World Health Organization (WHO), which indicate that men comprise approximately 10–15% of the global nursing workforce. This aligns with the Australian figure of approximately 10% and highlights the broader gender imbalance in nursing internationally. Regarding the statement about a decline in men in nursing, we acknowledge that while historical context was provided, specific figures were not included. We have revised the manuscript to clarify that the decline refers to the historical reduction in male representation in mental health nursing, particularly following sociocultural and educational shifts such as deinstitutionalisation and the feminisation of nursing education. Finally, while we did not use a formal methodology to guide the motivating factors for our qualitative findings, these were informed by existing literature and the thematic patterns that emerged during data analysis. We have clarified this in the revised manuscript to enhance transparency.
- Background: You indicate a pressing need for research on males in mental health (line 118). I wonder if there have been any studies in the past. You indicate some past issues, but do not highlight past research. I think a statement indicating poor research would connect this.
Thank you for this valuable observation. We agree that the background section would benefit from a clearer connection between the historical context and the current research gap. While there have been broader studies on gender in nursing and masculinity in healthcare, specific research focusing on the experiences of male mental health nurses remains limited. To address this, we have revised the section to explicitly acknowledge the lack of targeted research in this area.
- Problem Statement: Line 126 indicates the decline in male representation in mental health nursing, and again this needs to be explicitly stated with statistical figures. There has been a drop in working nurses – is this what you are referring to or is there other figures that support this?
Thank you for this important clarification. We acknowledge that the original statement lacked specific statistical support. While comprehensive longitudinal data on the decline of male representation in mental health nursing is limited, recent literature highlights a historical reduction in male participation in this specialty. A 2025 scoping review by Reedy et al. notes that men were once predominant in mental health settings, particularly during the institutional era, but their representation has declined significantly due to sociocultural changes and the feminisation of nursing education. We have revised the manuscript to reflect this more clearly and to distinguish this trend from the broader decline in the overall nursing workforce.
- Line 134-136 is not quite congruent with your aim, and I am not sure if it is intended to be your aim. If it is not, then I suggest revising the statement to be in alignment with your specific aim.
Thank you, for clarity, this has now been removed from the text
- Aim: Again, the commas around “who are male” make this a little unclear for me as described in the abstract section above. I wonder if in the Aim or Objectives, if it would be appropriate for you to put in a working definition of what “male” was for this study. You indicated in the findings that participants reported that patients had differing ideas about this. It would clarify for readers what you intended.
Thank you for your continued attention to the clarity and inclusivity of our language. In response to your feedback, we have added a dedicated statement at the beginning of the paper to clarify our use of gender terminology.
- Methods: Sample: Are you able to clarify if the recruitment was from one geographical area in Australia or not, and how researchers reached out? Also, clarify how the identification of being male was identified, how long participants had to identify as male, how long did they have to work as a mental health nurse, did they have to be male for the entire time they were mental health nurses, were there age restrictions?
Thank you we have provided additional data within the sample section of the paper
- Data Collection: For Microsoft Teams, can you address any data collection issues with this method – i.e., could cameras be turned off? And if so, was this a limitation? Also were there any threats to data with this method and how were protections offered?
Thank you for your thoughtful feedback regarding the use of Microsoft Teams for data collection. We acknowledge that online platforms can present challenges, including the potential for participants to turn off their cameras, which may affect group dynamics and non-verbal communication. To mitigate this, participants were asked to keep their cameras on throughout the session to support engagement and interaction, and all agreed to this request prior to the focus groups. We have clarified this in the revised manuscript.
Additionally, we have addressed data protection concerns by outlining the security measures in place. Microsoft Teams was selected for its compliance with institutional data security standards. All recordings were stored on encrypted, password-protected servers accessible only to the research team. These revisions have been incorporated into the data collection section to enhance transparency and address potential limitations.
- Data Analysis: Please clarify if researchers transcribed together or independently and how any discrepancies were resolved. Did you use a method from the literature for credibility?
This is stated already in the text, however, we have provided additional context to ensure this is explicit
- Results: For the table, is there any potential for there to be a threat to anonymity with the organization of data in this manner – i.e., can anyone else recognize participants from this information?
Thank you for this concern. We have ensured that this is not the case. Participants have agreed to the data in the table.
- Line 308-309 statement indicating that all participants had long standing careers in mental health nursing conflicts with your recruitment statement for your sample (line 170) indicating that you included early to mid-career nurses.
This statement has now been resolved for accuracy
- Discussion: For some of the discussion, I think the paper could benefit from describing some of the terminology in the introduction/background at the outset (lines 568-577) – compassionate performative or ethically masculinity for example…so that it is clear what your discussion points are emphasizing
Thank you for this helpful suggestion. We agree that introducing key terminology earlier in the manuscript enhances clarity and strengthens the reader’s understanding of the discussion. In response, we have added a brief explanation of the terms at the point of when they are used to assist the reader understand these more explicitly
Reviewer 4 Report
Comments and Suggestions for Authors
- The information about the study location, participants, and study design was not provided in the study title. Please improve it.
- The study’s novelty should be presented in the abstract section.
- The “nurse man” in the abstract and title was not clear. Please improve it.
- The issue of men in nursing was a lack. So this study lacks of scientific background, the problem statement regarding men in nursing, and also the novelty.
- The study location was also not discussed well. Why was the study conducted in a selected location? What makes it different from other countries?
- Overall, please improve the introduction section significantly.
Author Response
Reviewer 4
- The information about the study location, participants, and study design was not provided in the study title. Please improve it.
Thank you, we have now revised this.
- The study’s novelty should be presented in the abstract section.
Thank you, we have now revised this.
- The “nurse man” in the abstract and title was not clear. Please improve it.
Thank you for your helpful feedback. We acknowledge that the phrase “nurse man” in the abstract and title may have been unclear or potentially misleading. Our intention was to highlight the gendered experience of men in mental health nursing without reinforcing stereotypes or suggesting that men are fundamentally different from their peers. We have therefore revised the title and abstract to use clearer and more inclusive language, specifically referring to “men in mental health nursing” or “mental health nurses who are male”, to better reflect the focus of the study and ensure clarity for readers.
- The issue of men in nursing was a lack. So this study lacks of scientific background, the problem statement regarding men in nursing, and also the novelty.
This has been revised based on feedback from the other reviewers. We hope this has been addressed sufficiently
- The study location was also not discussed well. Why was the study conducted in a selected location? What makes it different from other countries?
This has been revised based on feedback from the other reviewers. We hope this has been addressed sufficiently
- Overall, please improve the introduction section significantly.
Thank you, we have received very positive feedback form the other reviewers regarding the introduction section, however, have made additional revisions to improve this section.
Round 2
Reviewer 4 Report
Comments and Suggestions for Authors
- The background section appropriately highlights the issue of career development among male nurses. However, it lacks a clearly articulated problem statement regarding the broader implications of the underrepresentation of male nurses on global health trends, particularly in Australia. It would be beneficial to explore whether the limited number of male nurses—especially in mental health services—has any correlation with adverse outcomes, such as increased patient aggression, higher staff injury rates, or even elevated morbidity or mortality due to suboptimal gender-sensitive care.
- Additionally, the manuscript misses an opportunity to position this issue within a global context. In some countries, it is recognized that male nurses are predominant in psychiatric facilities—particularly in male wards—due to the specific needs and safety considerations related to the patient population. This raises a potentially important gap in the current literature: to what extent have healthcare institutions across various countries integrated gender considerations into workforce planning, particularly in mental health settings? Addressing this question could strengthen the rationale and significance of the study (introduction section).
- The author also needs to promote “why” should be in the mental health?
- While the study presents valuable insights derived from interviews, it is unclear whether the authors employed any validation strategies, such as triangulation, member checking, or peer debriefing, to enhance the trustworthiness of the data. Clarifying whether such techniques were used would strengthen the methodological rigor and credibility of the findings.
Author Response
Comments and Suggestions for Authors
- The background section appropriately highlights the issue of career development among male nurses. However, it lacks a clearly articulated problem statement regarding the broader implications of the underrepresentation of male nurses on global health trends, particularly in Australia. It would be beneficial to explore whether the limited number of male nurses—especially in mental health services—has any correlation with adverse outcomes, such as increased patient aggression, higher staff injury rates, or even elevated morbidity or mortality due to suboptimal gender-sensitive care.
- Additionally, the manuscript misses an opportunity to position this issue within a global context. In some countries, it is recognized that male nurses are predominant in psychiatric facilities—particularly in male wards—due to the specific needs and safety considerations related to the patient population. This raises a potentially important gap in the current literature: to what extent have healthcare institutions across various countries integrated gender considerations into workforce planning, particularly in mental health settings? Addressing this question could strengthen the rationale and significance of the study (introduction section).
- The author also needs to promote “why” should be in the mental health?
Response to Reviewer Comment:
Thank you for your thoughtful feedback and for highlighting the importance of gender representation in mental health nursing. We appreciate your suggestion to explore the broader implications of male nurse underrepresentation, particularly in relation to patient outcomes and workforce planning.
However, we would like to respectfully clarify that the suggestion linking male nurses to the management of patient aggression may inadvertently reinforce outdated stereotypes about gender roles in mental health care. Having males because of security also potentially devalues the capacity among individual clinicians, in that it surreptitiously insinuates that males should be the ones to whom aggression is directed.
Contemporary mental health practice emphasises trauma-informed, relational, and de-escalation approaches that are not inherently tied to gender. While physical safety is a consideration in some settings, the contribution of male nurses extends far beyond this, encompassing therapeutic engagement, emotional support, and leadership, areas equally vital to patient care and recovery. To suggest otherwise would be short sighted.
We agree that gender-sensitive workforce planning is an important area for further exploration. In response, we have revised the introduction to more clearly articulate the significance of male nurse representation in mental health, framed within a broader and more contemporary understanding of men who provide mental health nursing care.
Comments and Suggestions for Authors
- While the study presents valuable insights derived from interviews, it is unclear whether the authors employed any validation strategies, such as triangulation, member checking, or peer debriefing, to enhance the trustworthiness of the data. Clarifying whether such techniques were used would strengthen the methodological rigor and credibility of the findings.
Response to Reviewer Comment:
We appreciate the reviewer’s thoughtful observation regarding the importance of validation strategies in qualitative research. To enhance the trustworthiness and credibility of our findings, we employed several validation techniques throughout the study. Specifically, member checking was conducted by inviting participants to review their transcripts for accuracy and completeness, allowing them to amend, delete, or add to their responses. Additionally, collaborative coding was undertaken by multiple researchers through regular meetings to ensure consistency and shared interpretation, which served as a form of peer debriefing. We also maintained reflexive journals and engaged in ongoing team discussions to monitor potential biases and support analytical rigor. This has already been outlined within the text.
While triangulation of data sources was not employed due to the focused nature of the study and the homogeneity of the participant group, we have clarified this in the revised manuscript and added a section to Data Analysis to explicitly outline the validation strategies used. We trust this addition strengthens the methodological transparency and rigor of the study.